# Use of a mechanistic growth model in evaluating post-restoration habitat quality for juvenile salmonids

**Carlos M. Polivka**[1]*, **Joseph R. Mihaljevic**[2¤], **Greg Dwyer**[2]

**1** Pacific Northwest Research Station, USDA Forest Service, Wenatchee, WA, United States of America,
**2** Department of Ecology and Evolution, University of Chicago, Chicago, IL, United States of America

¤ Current address: School of Informatics, Computing, and Cyber Systems, Northern Arizona University, Flagstaff, AZ, United States of America
* carlos.polivka@usda.gov

## Abstract

Individual growth data are useful in assessing relative habitat quality, but this approach is less common when evaluating the efficacy of habitat restoration. Furthermore, available models describing growth are infrequently combined with computational approaches capable of handling large data sets. We apply a mechanistic model to evaluate whether selection of restored habitat can affect individual growth. We used mark-recapture to collect size and growth data on sub-yearling Chinook salmon and steelhead in restored and unrestored habitat in five sampling years (2009, 2010, 2012, 2013, 2016). Modeling strategies differed for the two species: For Chinook, we compared growth patterns of individuals recaptured in restored habitat over 15-60 d with those not recaptured regardless of initial habitat at marking. For steelhead, we had enough recaptured fish in each habitat type to use the model to directly compare habitats. The model generated spatially explicit growth parameters describing size of fish over the growing season in restored vs. unrestored habitat. Model parameters showed benefits of restoration for both species, but that varied by year and time of season, consistent with known patterns of habitat partitioning among them. The model was also supported by direct measurement of growth rates in steelhead and by known patterns of spatio-temporal partitioning of habitat between these two species. Model parameters described not only the rate of growth, but the timing of size increases, and is spatially explicit, accounting for habitat differences, making it widely applicable across taxa. The model usually supported data on density differences among habitat types in Chinook, but only in a couple of cases in steelhead. Modeling growth can thus prevent overconfidence in distributional data, which are commonly used as the metric of restoration success.

## Introduction

Advancement in restoration ecology comes from improvement of methods of evaluating restoration treatments. The most common approach has been to rely on observations of relative

**Data Availability Statement:** All relevant data are within the paper and supporting information.

**Funding:** Early portions of this work (2009-2010) were funded by Bonneville Power Administration (Project No. 2003-017-00), and by the American

Recovery and Re-investment Act enacted by
President B. Obama. The later activities (2012-
2016) were funded by the U.S. Bureau of
Reclamation. JRM was funded by a US
Department of Agriculture (USDA) National
Institute of Food and Agriculture (NIFA)
Postdoctoral Fellowship (2014-67012-22272).

**Competing interests:** The authors have declared
that no competing interests exist.

density in restored habitat and unrestored habitat, but such data often have high uncertainty
[1] or show weak or no effects of restoration [2, 3]. Mechanistic model fitting has become a
vital tool in conservation biology [4], but the models often still rely heavily on population
numbers or densities and correlative relationships between density and environmental vari-
ables [5]. Because differences in habitat quality drive heterogeneity in growth and development
across individuals, leading to life-history variation and trait evolution, [6], measurement of
growth can provide a more robust indication of habitat quality than numbers or densities [7–
9], especially given that growth is often correlated with survival and reproduction [10, 11].
Growth rates and other life-history traits can thus be important indicators of restoration suc-
cess, but are used only rarely in this context [12, 13]. With the proliferation of habitat restora-
tion efforts in species conservation, the the use of mechanistic models to describe growth data
has the potential to expand the theoretical basis of restoration ecology [14].

Considerable effort is invested in the restoration of salmonid fish populations, particularly
in the Pacific Northwest, USA [15, 16]. Mechanistic models to address restoration efficacy are
often used at the whole-population scale, which may involve the estimation of demographic
parameters and survival through stages of the life cycle [17–19]. Although growth and develop-
ment are key features of salmonid life-cycle models, studies of restoration effectiveness only
occasionally quantify salmonid growth or survival [3, 20, 21], requiring that related parameters
in life cycle models be estimated from correlative data in earlier research. The extent to which
these life history traits are changed by restoration is nevertheless of high importance in the use
of life cycle models to predict and confirm population-level responses [17].

There is no shortage of models that quantify effects of habitat quality on growth, ranging
from calculation of suitability indices [22] to individual-based behavioral models [23]. Growth
models are also useful for providing descriptions of growth patterns when there are multiple
ecological inputs that are difficult to measure continuously throughout the growth interval
[24–27]. In cases when growth data have been collected, it should therefore be possible to com-
bine models and data to provide more accurate assessments of restoration efforts. Such an
approach could augment, and increase confidence in, distribution and abundance studies of
salmonids and across species. In contrast to demographic studies, what is often missing in
growth studies is the use of robust statistical methods of fitting the models to data. In growth
studies, comparison of growth rates between habitats is usually based only on mean growth
rates [28]. Moreover, when mechanistic models [26, 27] are fit to the data, they are usually still
sufficiently simple that they can be transformed to the point at which linear fitting methods
can be used [29]. This restriction on model complexity likely reduces our ability to make infer-
ences about growth. Modern methods of non-linear fitting therefore provide an opportunity
to quantify the effects of habitat restoration on growth in a statistically robust fashion.

Here we present the results of a multi-year study of growth in sub-yearling anadromous sal-
monids. We use our data to illustrate the usefulness of mechanistic growth models in restora-
tion ecology. We collected size data, as well as growth over time, via fish marked and
recaptured through the entire rearing season in each of five years to determine whether growth
of young-of-the-year Chinook and steelhead is improved in restored habitat. Because these
fish grow more or less continuously during the growing season, we were able to fit mechanistic
growth models to both types of data. We used Bayesian model fitting techniques to estimate
the parameters of our growth functions, and we compared growth rates between individuals
with different habitat selection behaviors. Salmonid growth rates are rarely constant; therefore,
we allowed for the possibility that growth-rate parameters would change over time [30, 31].
Another complication in this study system is that salmonids can be mobile [32] and the model
had to be applied differently to Chinook because they have low-to-zero recapture rates in unre-
stored habitat over time frames needed to measure growth [33]. Steelhead recaptures were

sufficient in both habitat types, so we were able to use raw growth data to determine the extent to which the model accurately reproduces growth rates in the field. We then compared our estimates of growth rates with previous observations of relative density [34] to determine the frequency with which increased abundance in restored habitat is associated with increased growth rates. The model demonstrates relatively consistently improved growth associated with restoration in Chinook, but that this was the case in only two study years for steelhead. Nevertheless, key model parameters describing growth rate were supported by steelhead raw growth data. Thus, our model has a strong application to evaluation of salmon conservation efforts.

## Materials and methods

### Study system

There are myriad approaches to river restoration, sometimes with more than one target species [15, 21]. A common technique is the enhancement of fish habitat by installing engineered log jams (ELJs) that facilitate the additional aggregation of natural wood and create pools that serve as primary habitat for target species. ELJs may also and stimulate production of macroinvertebrate prey resources for fish [35]. In the Pacific Northwest, USA, use of ELJs is a common conservation strategy for threatened and endangered salmonids [2, 15, 36]. In interior Northwest river basins, sub-yearling anadromous Chinook salmon (*Oncorhynchus tschawytscha*) grow and develop in streams for one year, and steelhead trout (*O. mykiss*) grow and develop in streams for 1-3 years, before migrating to the marine environment. Seaward migration occurs at the smolt stage and, in principle, ELJs should improve growth and survival to this stage. However, distribution patterns of salmonids vary spatio-temporally, and differ between species [37], exacerbating the difficulty of assessing restoration efficacy with distribution and abundance studies [21, 34, 38–40]. Both species studied here show short-term affinity for stream pools [41], but Chinook do not remain in pools without cover for a mark-recapture interval long enough to measure growth [33].

The in-stream habitat enhancement project studied here was implemented in the Entiat River sub-basin (49.6567˚ N, 120.2244˚ W), a tributary of the interior Columbia River. Restoration efforts in the Entiat River spanned the years 2008-2016, and during this period, ELJs were installed in multiple reaches of the river. Natural accumulation of wood in the lower 15-20 km of the river is infrequent due to heavy residential and agricultural land use in the immediate floodplain. Thus ELJs have the potential to not only enhance pools for rearing salmonids, but to trap and accumulate any natural wood present. We compared a single reach 4.5-4.9 km from the confluence with the Columbia River, that had pools enhanced by ELJs, with a reach that was 0.3 km further upstream and had no ELJs (Fig 1). The reach treated with EJS was the only one in the basin at the beginning of the study in 2009. ELJ-enhanced pools in that reach ("restored pools"; N = 11) were compared with pools in the other reach ("unrestored pools"; N = 11) that lacked natural log structures.

Both types of pools occurred on the stream margins where nearly all sub-yearling habitat use occurs [37]. Unrestored pools were 20-40% smaller and shallower than restored pools, but [34, 42] showed that the unrestored pools had a similar rate of increase in fish density with increasing area to the restored pools. We therefore concluded that the unrestored pools served as reasonable controls for the restored pools.

### Fish capture and marking

We collected growth data using mark and recapture of young-of-the-year Chinook and steelhead in restored and unrestored pools during the growing seasons of 2009, 2010, 2012, 2013 and 2016. In each set of pools, capture and recapture were carried out every 10-14 days

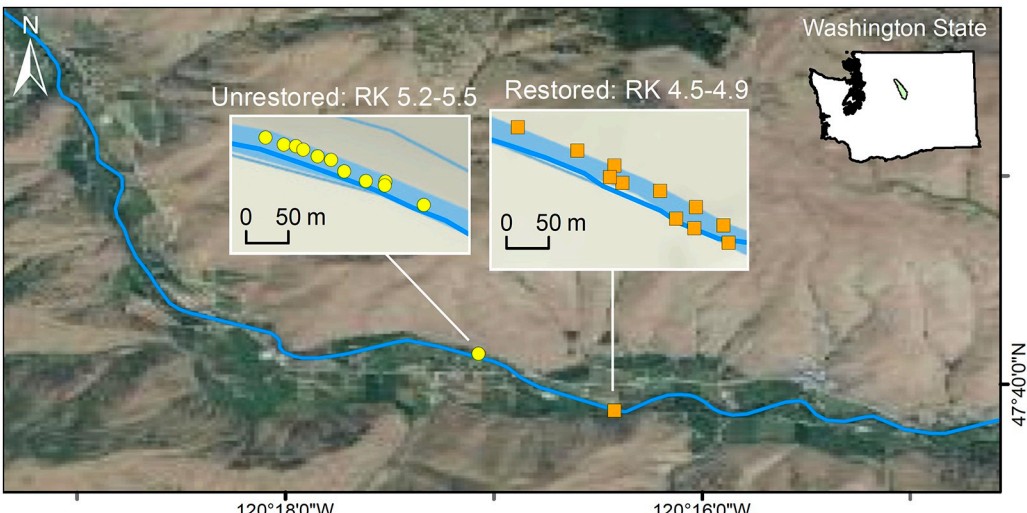

**Fig 1. Map of study area in the Entiat River.** Locations (river km relative to the confluence with the Columbia River) of the study reaches with restored pools and with unrestored pools are indicated.

throughout the growing season, and recapture intervals therefore ranged from 15-65 days [33]. We collected no growth data in 2011, 2014 or 2015 because periods of high water (2011) and post-fire sediment deposition (2014, 2015) prevented mark-recapture studies.

All fish handling was conducted under US Dept. of Commerce, NOAA-Fisheries Permit No. 1422, is consistent with the guidelines published by the American Fisheries Society. Fish were captured using a 3 m × 1.5 m × 3 mm mesh seine. The substrate and flow conditions made it impossible to pull the seine through the water, and so two field crew members instead stood at the downstream end of the pool and held the seine open as two other crew members, snorkeling in the water, used large hand nets to capture fish individually or to coerce fish into the seine. Visibility in the Entiat River is 4-5 m, so the two snorkelers could see the entire sampled area, ensuring that all visible fish were captured. Captured fish were transferred to insulated, aerated buckets for enumeration, marking and recording of size data (standard length, SL, in mm and mass in g).

Mild anaesthetization using MS-222 ($<0.1$ g · l$^{-1}$, 25% of the typical exposure tested for similar species [43]) exposure for 2-3 minutes made it possible to measure, weigh and apply an identifying mark to each fish. We avoided PIT tagging because it would have caused too much handling stress [44] to allow for robust mark and recapture, especially given the small size ($< 50$ mm SL) of some fish at the start of field sampling each year. We therefore marked fish with a subcutaneous injection of visual implant elastomer (VIE, Northwest Marine Inc.), a minimally invasive procedure. Individuals were identified by varying VIE color combinations and positions on the body. We marked $\sim 1000$ fish each year, with fewer than 2-3 deaths per year from handling/marking procedures. Following handling, fish were held in an aerated bucket for at least 10 min, or until they displayed a full righting response and normal activity, and were then released into the pool in which they were captured.

After 24 hrs field crews returned to the same pools as part of a separate short-term behavioral study [41] and captured all fish using the same methods. Recaptured fish were released to the pool and newly captured fish on that day were given marks as described above. The pools were sampled again 10-14 days later, with size data recorded on recaptured fish and marks applied to newly captured fish that had immigrated into the pool during the sampling interval.

## Data analysis

To analyze our data, we constructed mechanistic growth models that described fish growth over each growing season, and we fit these models to data on fish size. Size data included observations both of individuals marked and recaptured, and of individuals that were marked but never recaptured. We then compared the parameters of best-fit growth models for fish that used ELJ-enhanced pools with fish that did not. In these species, individual growth data are typically collected via mark-recapture studies, but the high mobility of Chinook [45] often limits their recapture rates, especially in unrestored habitats [33], and fitting growth models allowed us to nevertheless use these data. To do this, we combined the repeated size measurements on the recaptured individuals with individuals captured only once, regardless of the habitat of initial capture. The repeated measurements of recaptured fish ("recaptures") required use of a random effect in the model specification (see below), whereas the individuals captured only once ("others") only required fitting the model to the sizes observed during the course of the season. This analysis for Chinook is more conservative, because at least some of the "others" occupied restored habitat for some portion of the interval of time between sampling events. If there is a benefit to restoration, some portion of the "others" will therefore be more comparable to "recaptures," and may affect the likelihood of detection of growth differences with the model. For steelhead in contrast, we had a large number of recaptures in unrestored habitat (N = 179), and so we fit models to the data only for recaptured fish and compared habitat types.

Covariates including physical habitat characteristics (depth, current velocity, temperature) and biotic mechanisms (prey delivery, fish density, intra- and interspecific competition) are likely to interact with growth and with each other. These inputs are difficult to measure continuously throughout the growth intervals of individual fish, both because they could be marked and recaptured at different beginning and end points in the season, and because the length of the growth interval varied. Therefore, our model does not include parameters that consider each of these effects, but rather describes growth as the result of their combined effect. We do, however, address density in the context of comparing whether growth differences among habitat types are linked with density differences (see below). In this manner, our analyses provide a description of the effect of restoration on growth of individuals as a means of generating hypotheses about specific mechanisms.

**Model specification.**   We constructed our growth model to describe the growth of young-of-the-year Chinook salmon and steelhead in streams during the growing season. Size-over-time models in fisheries, e.g., [24] usually describe growth over the lifetime of the organism. One important consideration was that inspection of our data showed that there were inflection points; therefore, we used a generalized version of a logistic growth model [25–27]. This model allows for accelerating growth in the early stages of the growing season, followed by an inflection point, after which growth decelerates, leading to a maximum size at the end of the first season. The use of a size asymptote is consistent with patterns evident in our data, and in size data reported from several years of downstream migration of smolts in the spring [46]. Our model is:

$$l_t = Y' + \frac{L_\infty - Y'}{1 + e^{\hat{\alpha} - at}}. \tag{1}$$

Here, $l_t$ is the length at time $t$, $Y'$ is the lower bound on size, effectively a minimum, and $a$ is the rate of increase in the size curve. $L_\infty$ is the maximum size that a fish can reach during the growing season, so that $Y' \leq l_t \leq L_\infty$ for all time $t$. The $\hat{\alpha}$ parameter describes the timing of the

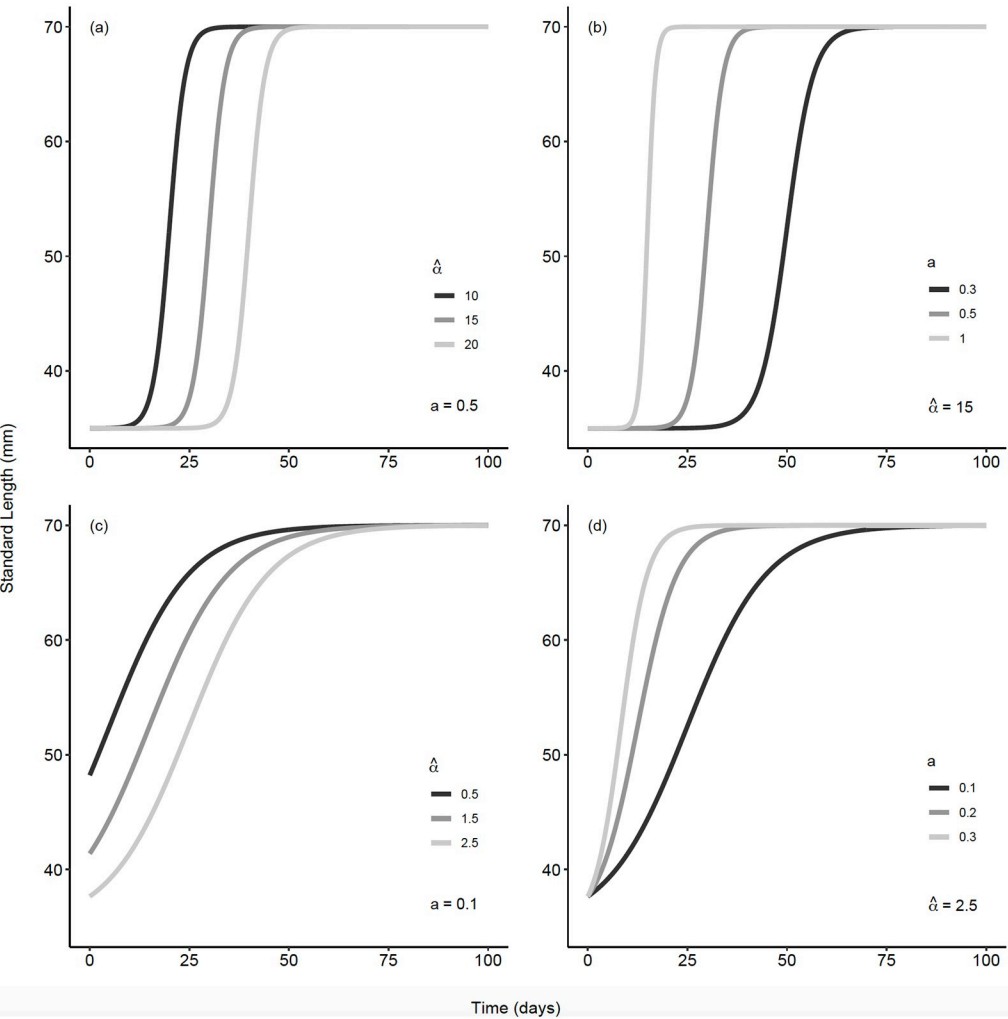

**Fig 2. Growth curves generated from Eq 1. over a 100 day growing period.** A: Effect of $\hat{\alpha}$ on the timing of the inflection point. B: Effect of $a$ on the slope of the growth curve. Panels C and D: Growth curves for values of the two parameters that are closer to those estimated from our data. For the curves in C in particular, the inflection point occurs close to time $t = 0$, so that the intercepts vary between parameter values. In these cases, fish are close to the lower bound on the curve $Y'$ when time is less than zero, which is to say, before sampling begins. In all panels $Y' = Y_\infty = 35$.

increase indicated by $a$ and contributes to setting the inflection point of the curve, which occurs at $\hat{t} = \frac{\hat{\alpha}}{a}$.

Importantly, both $\hat{\alpha}$ and $a$ can have profound effects on the dynamics during the growing season, potentially resulting in substantial differences in $l_t$ at any given time $t$. As the value of $\hat{\alpha}$ decreases, with the other parameters held constant, fish reach larger sizes earlier in the season. The $a$ parameter describes the steepness of of the curve at small sizes, so that increasing values of $a$ result in more rapid increases in size over time (Fig 2).

To avoid confusion with other models such as the Von Bertalanffy model, we re-parameterized the model according to $Y_\infty = L_\infty - Y'$, so that $Y_\infty$ is the total amount by which a fish increases in size during the growing season. The model is then:

$$l_t = Y' + \frac{Y_\infty}{1 + e^{\hat{\alpha} - at}}, \tag{2}$$

When non-linear models are fit to data, a standard approach is to transform the model into a linear form to accommodate linear least-squares fitting routines and their assumptions, such as normally distributed residuals. For our data, however, it turned out that the residuals are normal even without transformation of the model, and the variance in the residuals for the untransformed model was roughly constant. We therefore did not transform the model. Instead we simply added residual variation to the non-linear model, according to:

$$l_{it} = Y' + \frac{Y_\infty}{1 + e^{\hat{\alpha} - at}} + \epsilon_{it}, \tag{3}$$

Here $i$ indicates individual fish $i$. We then assume that the residual $\epsilon_{it}$ for fish $i$ at time $t$ follows a normal distribution with mean zero, and variance $\sigma^2_{residual}$, which we estimated from the data. We thus have $\epsilon_{i,t} \sim N(0, \sigma^2_{residual})$, so that $\sigma^2_{residual}$ is the residual variation in length. Thus, $Y'$ and $Y_\infty$ describe growth bounds averaged over individuals in the population. We then extended the model to allow for fixed treatment effects, and random year and individual effects. First, to include habitat treatment effects, restored vs. unrestored, we allowed the four model parameters ($Y'$, $Y_\infty$, $\hat{\alpha}$, and $a$) to vary by habitat type, as fixed effects. Second, to account for the statistical properties of our sampling design, we included random effects of year, to allow for annual variability in the parameters. Third, in the case of individuals that were recaptured during a season, we accounted for repeated measurements by including random effects of individual. Given these considerations, the model becomes:

$$l_{ihyt} = (Y'_h + \epsilon^1_{hy}) + \frac{(Y_{\infty h} + \epsilon^2_{hy})}{1 + e^{[(\alpha_h + \epsilon^3_{hy}) - (a_h + \epsilon^4_{hy})t]}} + \epsilon^5_{ih} + \epsilon^6_{iht} \tag{4}$$

Here $i$ is again the individual, $h$ is the habitat type, restored or unrestored, $y$ is the sampling year, and $t$ is time within a growing season. The $\epsilon$'s represent the different types of random effects, as follows.

The random effects of year on each of the four model parameters, $\epsilon^{1...4}_{hy}$, vary between the two habitats, such that, for example, $\epsilon^1_{hy} \sim N(0, \sigma^2_{Y'})$, where $\sigma^2_{Y'}$ is the random variation in initial size $Y'$ measured across habitats $h$ and years $y$. For the random effect of recaptured individuals, $\epsilon^5_{ih} \sim N(0, \sigma^2_{individual_h})$, where $\sigma^2_{individual_h}$ is the variance in fish length among recaptured individuals from a given habitat type. This random effect of individual affects only the overall length, rather than the rate of growth. Among other things, this means that we can account for the possibility of observing fish that start off larger than average being more likely to continue being larger than average throughout the growth period [47]. To be conservative, we began by assuming that no Chinook in the unrestored pools were recaptured, and so the random effects of individual Chinook are only relevant for recaptures in the restored habitats.

Finally, the residual variation is allowed to differ between the two habitats, where $\epsilon^6_{ht} \sim N(0, \sigma^2_{residual_h})$. This is particularly important for the Chinook data set, because some of the variation among the recaptured individuals in restored habitats can be explained by the random individual effects. In the unrestored habitats in contrast, no individuals were recaptured, and therefore more variation was left unexplained. Because of this, it turned out to be the case that $\sigma^2_{residual_{unrestored}} > \sigma^2_{residual_{restored}}$.

**Model fitting.**   We fit our models using Hamiltonian Monte Carlo (HMC) with the open-source software, *Stan* [48] and **R** [49], utilizing the package *rstan* [50]. The HMC algorithm provides an efficient method of fitting nonlinear models, and the software platforms ensure that our methods are open and reproducible (see Supporting Information).

In general, we used vague priors for our parameters, but to aid HMC performance, we centered the prior distributions of $Y'$ and $Y_\infty$ on realistic values, based on previous work with these two species. Also, we constrained $\hat{\alpha}$ and $a$ to fall within realistic ranges, namely 0-10 and 0-1, respectively, again based on previous work. Sensitivity analysis showed that these priors did not strongly influence our posterior inferences, and that the posterior was clearly dominated by the likelihood, rather than by the prior distributions. For each model, we ran three HMC chains for 9000 iterations, using the first 2000 iterations as a warm up. We then thinned by 7 iterations to produce a total of 1000 samples per chain. We evaluated HMC chain convergence based on Gelman and Rubin's potential scale reduction factor, $\hat{R}$ [51]. We visually inspected chain mixing using traceplots, running mean plots, and marginal posterior density plots [51], and we saw no evidence of pathological MCMC behavior. Although we thinned our chains to avoid auto-correlation, our effective sample size was close to the total number of iterations, and thinning was likely unnecessary [52].

To compare parameter estimates between the restored and unrestored habitats, we calculated differences between paired parameter values for the restored and unrestored habitats for each sample in the model's joint posterior. For example, to calculate differences in the shape parameter between restored habitats $r$ and unrestored habitats $u$, we calculated $\hat{\alpha}_r - \hat{\alpha}_u$ for each of the 3000 samples in the posterior (our 3 Markov chains each produced 1000 samples). We then calculated the 95% credible interval (CI) of that set of differences. This approach allowed for the possibility of correlations in parameters across the samples in the posterior, and is therefore a more robust method than calculating parameter differences by making independent draws of each parameter from the posterior [53].

Note that, in the case of Chinook, the $u$ group consisted of all uncategorized fish, meaning transient individuals that were not recaptured, regardless of habitat of original capture. Accordingly, when we tested for interactions between habitat and year for Chinook, we took into account the random year effects. For example, in testing for differences in the shape parameter between years for Chinook, we calculated $(\hat{\alpha}_r + \epsilon^1_{r,y}) - (\hat{\alpha}_u + \epsilon^1_{u,y})$ for each posterior draw, where $\epsilon^1_{r,y}$ and $\epsilon^1_{u,y}$ are the random year effects in year $y$.

We considered the parameter estimates for the two habitats to be meaningfully different if the 95% CIs of their differences did not overlap zero. For most parameters, however, the fraction of differences that was above or below zero was reasonably consistent across years, even if the 95% CI overlapped zero. Because this consistency provides at least modest additional support for some of our arguments, we report the fraction of differences that were above or below zero in each case.

Our best-fit models suggested that fish almost always exceeded the minimum size $Y'$ before sampling began. To estimate fish size at the beginning of sampling, we therefore used the best-fit models to back calculate median fish length at time $t = 0$ when observations began. This also allowed us to confirm that there were no differences in fish size among habitat types at the start of the season. Such initial differences can lead to growth advantages that are independent of environmental factors such as habitat quality [47].

**Model comparison with direct measurement of growth in Steelhead.** Because we recaptured steelhead in both restored and unrestored pools, we were able to directly compare observed growth rates for recaptured steelhead between habitat types, in addition to comparing our estimates of growth that came from fitting models to the data. In carrying out analysis of measured growth rates, a key consideration is that, in many organisms, individual growth rates are strongly size dependent [47, 54, 55]. Because we encountered a wide range of initial sizes (40-75 mm SL), it was important to allow for the possibility of size-dependent growth when comparing steelhead growth rates between habitats.

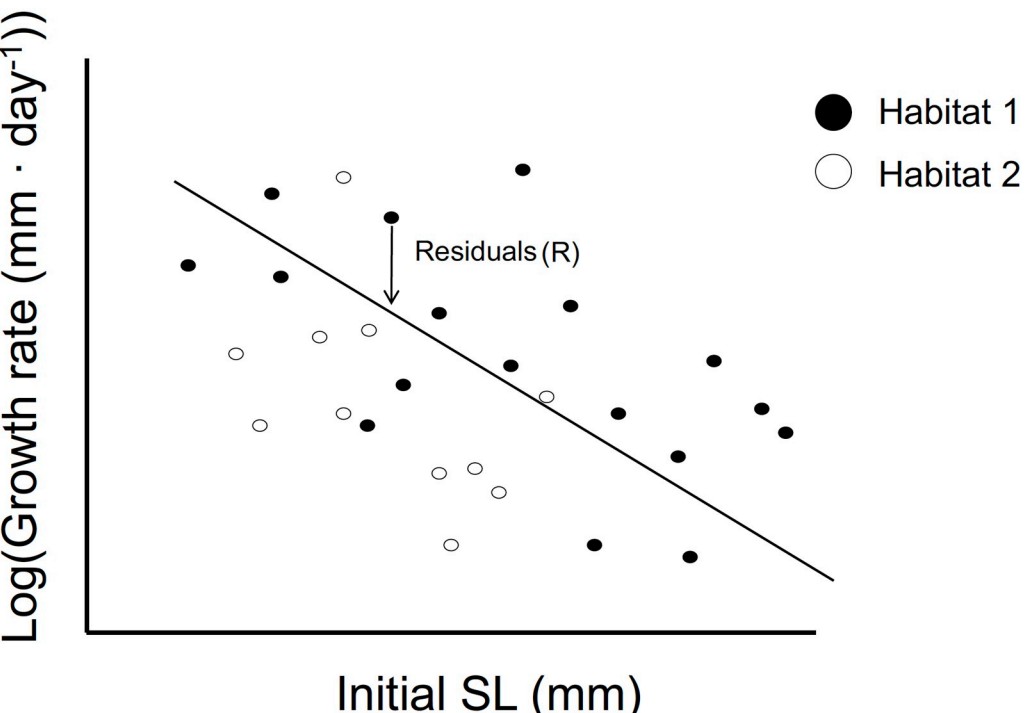

**Fig 3. Use of residual values (R) from a size vs. growth rate regression to compare growth in different habitats.** The arrow shows the distance from the points to the line. Mean residual values $\bar{R}$ are calculated for all individuals in each habitat (r = restored, u = unrestored), and if $\bar{R}_r > \bar{R}_u$, then we conclude that growth is higher in restored than in unrestored habitats.

We generated a predicted growth vs. size regression for steelhead for each year, by carrying out a log-linear regression of change in size ($mm \cdot day^{-1} = \frac{SL_{recaptured} - SL_{marked}}{days}$) on initial standard length, for each recaptured individual. To compare growth in restored vs. unrestored habitat in these regressions, we kept track of the habitat in which each individual was recaptured. We then used the residual for each point as an indicator of growth rate relative to the population average for that size, where the average for the size was calculated from the regression line of growth on size. A positive residual indicates that an individual has a higher than average growth rate, adjusted for size, whereas a negative residual indicates that an individual has a lower than average growth rate, again adjusted for size (Fig 3). In comparing size-dependent growth rates between habitats, we compared the mean residual from restored habitat to the mean residual from unrestored habitat for each year of the study, using two-sample *t*-tests. Because growth rate in $mm \cdot day^{-1}$ corresponds to the *a* term in Eqs 1–4, we compared differences among habitats in mean residuals with differences among habitats in the *a* parameter.

**Concordance of growth and habitat occupancy patterns.** From a management perspective, our goal was to determine whether observations of density differences between habitats are supported by growth differences. We therefore compared our model output first to the fish density data from [34] for each of the study study years 2009, 2010, 2012, and 2013. For 2016, we had density data available from initial captures during these mark-recapture studies and made density comparisons using the same methods as in [34].

## Results

### Chinook salmon

Across the five sampling years (2009, 2010, 2012, 2013, and 2016), we collected 3,568 length records from Chinook. In the ELJ-enhanced pools, we recaptured 238 Chinook, for a total of 481 length records, with five fish that were recaptured more than once. The best-fit growth curves in Fig 4 show that those recaptured Chinook were generally larger early in the season compared to unrecaptured fish. This visual impression is confirmed by a lower value of $\hat{\alpha}$ for individuals recaptured in ELJ pools compared to $\hat{\alpha}$ values for unrecaptured fish (Fig 5). Because $\hat{\alpha}$ is an inverse measure of size early in the season, the difference in $\hat{\alpha}$ values means that the recaptured fish were larger than unrecaptured fish early in the season. Negative values of the difference in $\hat{\alpha}$ values occurred in 98.5% of sample differences for all years combined, and in 98.3%-100% of samples for 2009, 2010, and 2013 (Fig 5). In 2012 and 2016, the 95% CIs overlapped zero, but negative values occurred in > 90% of samples, consistent with the differences for the other years and with the differences for all years together.

In 2009 and 2010, however, values of the growth rate $a$ were larger for unrecaptured fish than for recaptured fish. This observation, in combination with the larger $\hat{\alpha}$ values for unrecaptured fish describing small early-season size, indicates that unrecaptured fish experienced a rapid increase in size later in the season (Fig 4) that compensated for the early season disadvantage. The difference in those two years was indicated in >99% of differences in sample values drawn from the posterior distributions of $a$ (Fig 5). Although the 95% CI for the differences in $a$ for all years together overlapped zero, they were below zero in 91.1% of the draws, consistent with the trend observed in 2009 and 2010.

All else equal, smaller $\hat{\alpha}$ values lead to earlier inflection points $\hat{\alpha}/a$. To test whether the smaller values of $\hat{\alpha}$ for Chinook recaptured in restored habitat in all years combined, and in 2009, 2010, and 2013, led to earlier inflection points for those fish, we drew pairs of values of the shape parameter $\hat{\alpha}$ and the growth parameter $a$ for recaptured and unrecaptured Chinook. We then calculated differences in inflection points $\hat{\alpha}/a$ for recaptured and unrecaptured fish in each year and for all years combined, as we did for all the individual parameters.

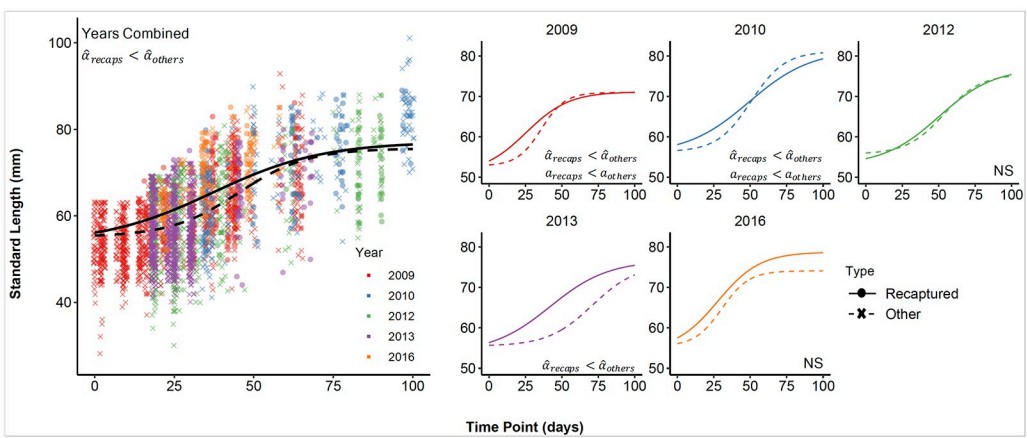

**Fig 4. Growth curves based on estimated model parameters (mean of the posterior) for Chinook salmon recaptured in treated habitat ("recaps," solid lines) vs. those captured in either habitat type, but not recaptured ("others," dashed lines).** The left panel shows the data and the habitat-specific models for all years combined. The small panels to the right show the data and the habitat-specific models for individual years. Parameter differences (see Fig 5) are shown in the upper left corner of each panel.

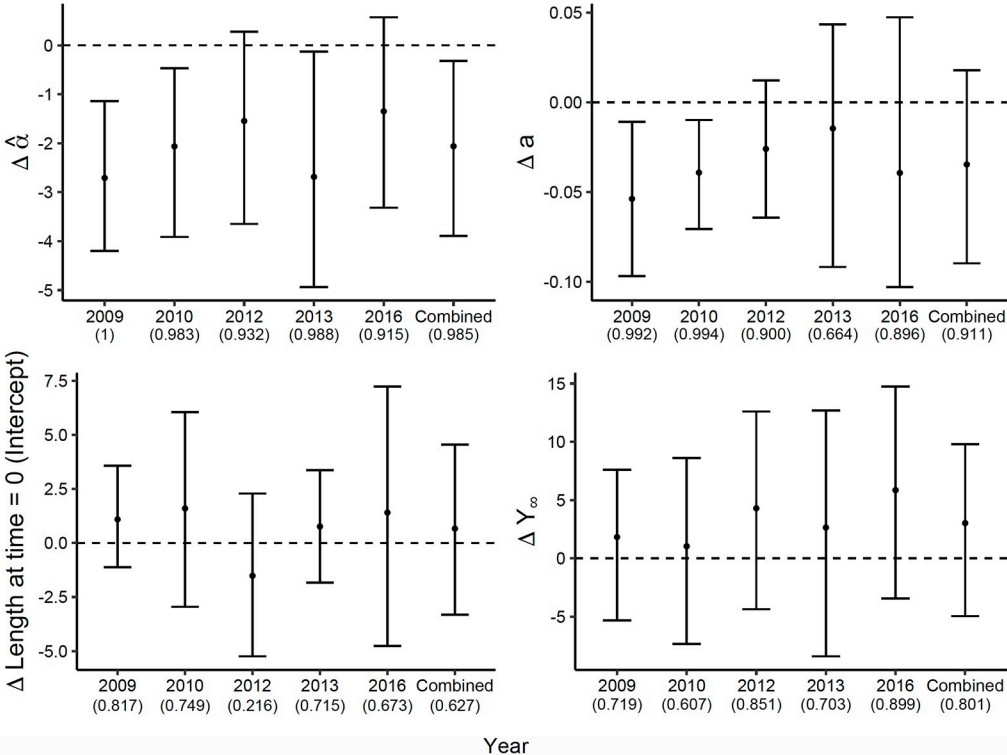

**Fig 5. The difference (Δ; recaptures − others) in estimates of all model parameters ± the 95% credible interval between the two capture types of young-of-the-year Chinook salmon.** Recaptures in restored habitat; others = fish marked and not recaptured, regardless of capture origin. Year-specific estimates incorporate the random effect of year, and combined differences are based on the average parameter values among years. The number in parentheses represents the fraction of posterior draws for which the difference was above or below zero, as appropriate. Δ Length at time = 0 was substituted for model parameter $Y'$.

This procedure showed that inflection points for recaptured fish did indeed occur earlier than for recaptured fish in 2009, with 99.8% of sampled differences being less than zero. In 2013, there was a similar trend, with 94.6% of sampled differences less than zero. Differences in the other years, and for the combined data, were not meaningful, likely because independent variation in the two parameters obscured differences in the inflection points. We therefore conclude that there is at least modest evidence that restoration is associated with larger Chinook size earlier in the season as indicated by the shape parameter $\hat{\alpha}$, but that variation in the two parameters prevents a consistently earlier inflection point.

## Steelhead

We recorded the lengths of 2,871 steelhead across the five sampling years. In the ELJ-enhanced pools, we recaptured 306 individuals for a total of 689 observations, whereas in the unrestored habitats, we recaptured 179 individuals for a total of 413 observations. The number of steelhead recaptured from unrestored habitats was thus sufficient to fit the model directly to recaptured individuals in each habitat type, in contrast with Chinook. In 2013, however, there were no recaptures of steelhead in unrestored habitat for recapture intervals of longer than 24h. We therefore excluded the 2013 data from our analyses.

For all years combined, there were no differences in growth parameters in steelhead recaptured in restored versus unrestored habitats (Fig 6), but there were differences in some

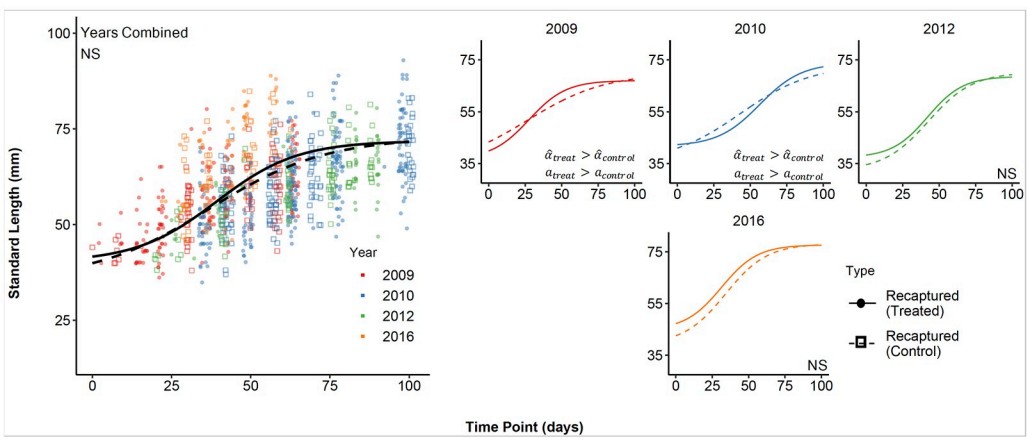

**Fig 6. Growth curves based on estimated model parameters (mean of the posterior) for steelhead recaptured in restored habitat (solid lines) vs. those recaptured in unrestored habitat (dashed lines).** Left panel shows data and fitted curves for all years combined; additional figures show individual years. Parameter differences (see Fig 7) are indicated in the lower right corner of each panel. The analysis omits 2013 owing to lack of long-term steelhead recaptures in unrestored habitats.

parameters in 2009 and 2010 (Fig 7). In those two years, both the shape parameter $\hat{\alpha}$ and the growth rate parameter $a$ were higher for fish in restored habitat in 99-100% of samples from the posterior distributions. Over all years, almost 90% of differences in $\hat{\alpha}$ were above zero, and almost 93% of differences in $a$ were above zero, supporting the meaningful differences seen in 2009 and 2010.

The higher values of $a$ for fish in ELJ pools mean that growth rates were higher for steelhead in those two years, but the higher values of $\hat{\alpha}$ mean that fish in restored habitat reached larger size later in the season and caught up to the fish in unrestored habitats that were larger earlier. The lack of differences in total growth $Y_{\infty}$ supports this conclusion. The inflection point was later for steelhead in restored habitats in 2010 only (99.3% of sampled differences above zero), again suggesting that the inflection point is not a specific indicator of growth differences among habitats. Nevertheless, mid-season occupancy of restored pools was associated with rapid growth.

## Direct observations of growth in Steelhead

Residuals from the log-linear regression of growth (mm · day$^{-1}$) on initial size were more positive, on average, for individuals marked and recaptured in restored habitat than in unrestored habitat, indicating higher growth in unrestored habitat. This trend, however, held for 2009 and 2010 but not 2012 or 2016 (Fig 8). Inspection of the regression plots for 2009 and 2010 indicated that the largest differences in residuals between habitats in those two years were among individuals 55-60 mm in length. Separate residual analyses on individuals < 60 mm, and individuals > 60mm, produced significant differences in residuals only in individuals < 60 mm (not shown).

## Concordance of growth and density

In each year of our study, Chinook salmon were more abundant in restored habitat than in unrestored habitat whereas parameters from the growth model only showed a growth benefit in three of the five years (2009, 2010 and 2013; Table 1). For all years combined, however, density and growth were concordant for Chinook. For steelhead, 2009 was the only year in which

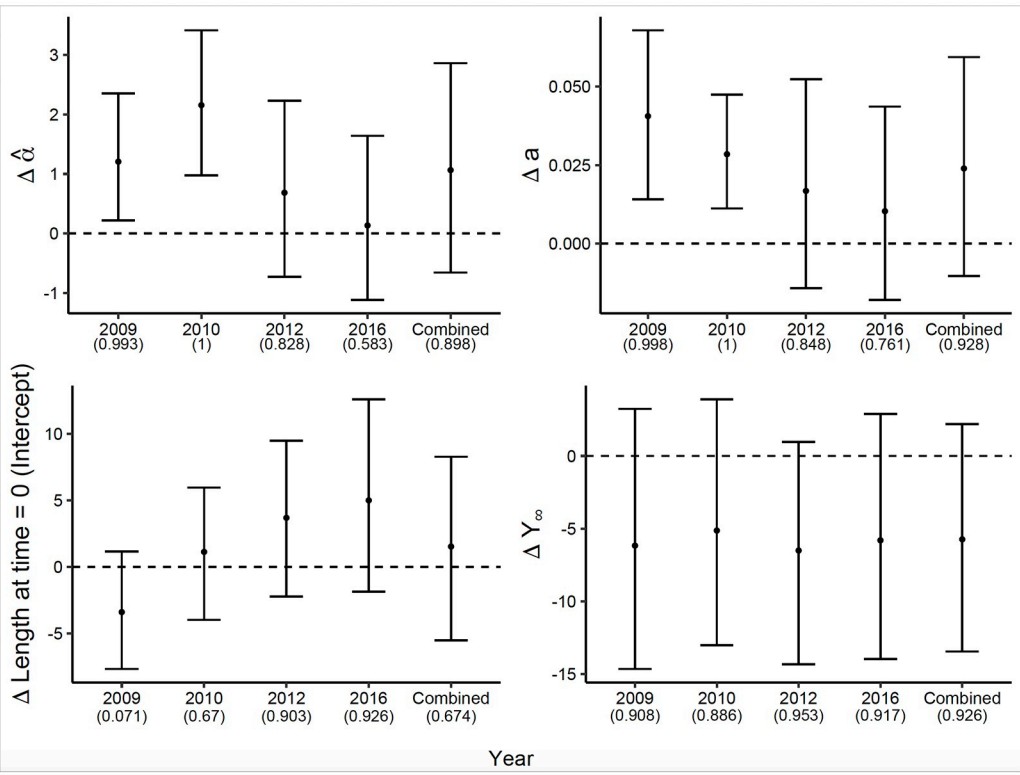

**Fig 7. The difference in estimates of all model parameters ± the 95% credible interval between the two capture types of young-of-the-year steelhead (recaptures in restored habitat—recaptures in unrestored habitat).** Year-specific estimates incorporate the random effect of year, and combined differences are based on the average parameter values among years. The number in parentheses represents the fraction of posterior draws for which the difference was above or below zero, depending on trend of the data. For all panels except the lower right, this represents the fraction of posterior draws above zero. $\Delta$ Length at time = 0 was substituted for model parameter $Y'$. The analysis omits 2013 owing to lack of long-term steelhead recaptures in unrestored habitats.

there was concordance between observed fish abundance and growth patterns. In 2010, when there was no significant difference in density among habitats, there was nevertheless more rapid growth in restored pools, as indicated above by the *a* term in the model and by the positive mean residuals from the growth vs. size curve (Fig 8 and Table 1).

## Discussion

Mechanistic ecological models play an important role in restoration ecology [56], because their description of observed data increases the chances that species conservation efforts will be successful. Perhaps the best known set of tools comes from population viability analysis, widely used in the development of recovery criteria, and in calculations of the amount of habitat that needs to be restored [57]. Species distribution analysis is a related approach that is used to identify the habitat characteristics that are important for conservation [58, 59]. Like population viability analysis, species distribution analysis is used to predict how species respond to management actions or environmental change [60]. Predator-prey theory has been used to construct more complex models, again to analyze the outcomes of conservation actions [61].

All these methods, however, assume that there are correlations between abundance and habitat characteristics, but the support for such correlations is often mixed [62]. Growth data in contrast are infrequently used in restoration ecology. Surveys of relative abundance cannot

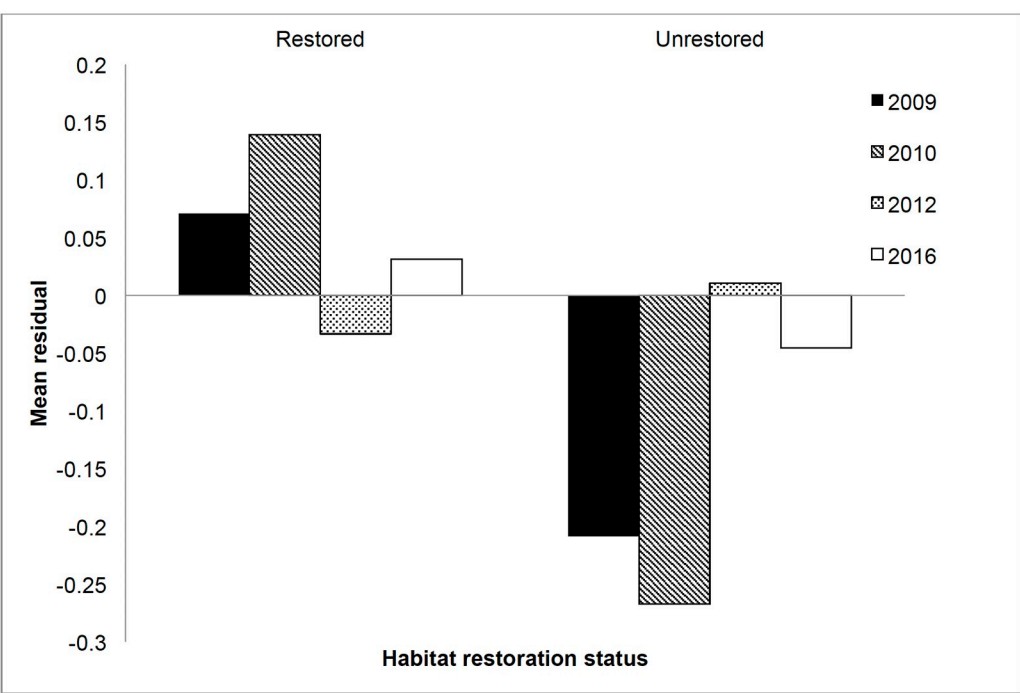

**Fig 8. Mean residuals from a log-linear regression of growth rate versus size, in restored vs. unrestored habitat, for each study year.** The analysis omits 2013 owing to lack of long-term steelhead recaptures in unrestored habitats.

easily identify the processes that determine ecological differences between habitat types [7, 63, 64]. Data sets on growth thus provide opportunities to expand the use of mechanistic models in describing specific effects of habitat restoration in conservation efforts. By comparing growth between restored and unrestored habitat, we have shown that restoration enhances

**Table 1. Concordance of occupancy differences and growth-parameter differences.** Recall that large $\hat{\alpha}$ means that the inflection point comes later in the season, indicating less favorable growth conditions (NS = No difference/not significant).

| Year | Density higher in . . . | Growth more favorable in. . . | Parameters |
|---|---|---|---|
| *a) Chinook Salmon* | | | |
| 2009 | Restored | Restored | $\hat{\alpha}_{restored} < \hat{\alpha}_{unrestored}$ |
| | | | $a_{restored} < a_{unrestored}$ |
| 2010 | Restored | Restored | $\hat{\alpha}_{restored} < \hat{\alpha}_{unrestored}$ |
| | | | $a_{restored} < a_{unrestored}$ |
| 2012 | Restored | NS | |
| 2013 | Restored | Restored | $\hat{\alpha}_{restored} < \hat{\alpha}_{unrestored}$ |
| 2016 | Restored | NS | |
| **Years Combined** | Restored | Restored | $\hat{\alpha}_{restored} < \hat{\alpha}_{unrestored}$ |
| *b) Steelhead* | | | |
| 2009 | Restored | Restored | $a_{restored} > a_{unrestored}$ |
| 2010 | NS | Restored | $a_{restored} > a_{unrestored}$ |
| 2012 | Restored | NS | |
| 2013 | NS | NS | |
| 2016 | NS | NS | |
| **Years Combined** | Restored | NS | |

growth rather than simply redistributing organisms among habitats or increasing numbers. Compared to observations of abundance, fitting mechanistic growth models to data can thus provide stronger evidence in favor of restoration.

Growth is an important life history trait because of its correlation with fitness in many species [10, 11], including salmonids [65]. Growth models can therefore help identify the potential fitness advantages of a given habitat type. Accordingly, although we designed our analysis to focus on a specific study system, our over-arching goal is to to provide an adaptable tool for a wide range of studies. Moreover, by using state-of-the-art Bayesian methods, we have shown how growth models can be used to make statistically robust inferences in restoration ecology. The development of Bayesian analytical methods has similarly allowed for the expansion of the applicability of ecological models in other areas of restoration ecology. For species distribution analysis, for example, Bayesian methods have provided useful examples of the potential of advanced statistical computing in restoration ecology [66–68].

Although our model describes a specific life history trait for our study species, in the form of growth during early development, we constructed our model by adapting models that describe growth over the lifetime of an organism. Our model could thus be expanded to describe longer segments of a species' life history. Moreover, our model offers two advantages over previous models. First, our model includes parameters describing not only the rate of growth, represented by $a$, but also the timing of size increases, represented by the shape parameter, $\hat{\alpha}$. Second, the model allows for spatial variation, in that its parameters and residuals were allowed to vary by habitat type, thus allowing for direct description of habitat differences in the data. Finally, our model allows for the high mobility that can be observed in stream salmonids [32]. For Chinook salmon, the comparison was between recaptured individuals and individuals that were only captured once. This use of a time series of size data as indicative of growth trajectories is not new but, combined with the complexity built into our model and our analytical technique can be applied to many species with a range of mobility.

Our estimates of the model parameters for sub-yearling Chinook salmon and steelhead demonstrate that habitat restoration can make growth conditions more favorable. Our estimates of the shape parameter $\hat{\alpha}$ showed that Chinook reached large size earlier in three of the five study years, and across all years combined, as a result of remaining in ELJ-enhanced pools. In two of the study years, the growth rate parameter, $a$, was higher among individuals that moved among habitat types, but this occurred later in the season. The interpretation of model parameters therefore can be specific to the study system.

After growth to the parr stage (late August/early September), Chinook begin migrating downstream to overwintering habitat [45, 69]. As Chinook migrate downstream, slower growing, later hatching, or previously more transient Chinook can immigrate into the pools being vacated by earlier occupants. This can lead to compensatory growth because of release from competition [70]. Chinook previously at a growth disadvantage may then increase their growth once their competitors leave, and may ultimately grow faster than their competitors once did [71]. This effect also likely depends on the overall population size and distributional differences between habitat types. The largest difference in Chinook abundance between habitats was in 2009 and 2010 [34]. Thus, fish in those two years were more likely to have been originally captured in ELJ pools. By mid-season, individuals captured in those pools could have occupied a pool for up to 10-14 days, the typical interval between sampling events, even if not recaptured later. This could explain the higher growth rate $a$ in unrecaptured Chinook initially captured during mid-season of 2009 and 2010 (Fig 5).

As an explanation for observed habitat differences in steelhead growth in 2009 and 2010, competitive release is also consistent with estimated growth parameters. In this study system, steelhead are less abundant overall and overlap with Chinook, particularly in ELJ pools [34].

When steelhead are less abundant than other salmonids, they usually respond by adopting a more generalist habitat use pattern [72]. Indeed, steelhead recapture rates in unrestored pools indicate higher residency time there compared with Chinook [33], despite the fact that they are usually relatively more abundant in ELJ pools [34, 73]. If early season growth by steelhead in ELJ pools is slowed by competition with Chinook, then steelhead might do better in unrestored pools, especially in early season.

Moreover, the competitive pressure exerted by Chinook is probably more severe when Chinook are at high density. Further evidence that competition affects steelhead growth then comes from the observation that the $\hat{\alpha}$ parameter for steelhead was larger in restored habitat in 2009 and 2010, when Chinook densities were highest. In those two years, the $\hat{\alpha}$ parameter indicated that steelhead attained larger size earlier in unrestored habitat. The opportunity to grow rapidly in restored habitat with less competition becomes available mid-season when Chinook begin downstream migration. Because our best-fit models also showed that transient Chinook showed higher growth during this time, our model-fitting approach has apparently revealed the effects of both intra- and interspecific competition.

Direct measurements of growth in in steelhead supported the inferences we made by model-fitting. Although direct measurements can in some cases provide more obvious answers, our model-fitting approach has important advantages. First, our model-fitting approach is capable of not only comparing habitats directly, but also of using time series of size to compare recaptured and non-recaptured individuals (as in the case of Chinook). Thus it also integrates over the effects of environmental variables, which are difficult to include as individual correlates in other approaches, across the whole rearing season. Second, model-fitting allowed us to test for effects of different habitats on Chinook, even though the Chinook recapture rate in unrestored habitat was nearly zero [33]. Third, our approach demonstrated that subtle differences in growth exist between early- and mid-season for both species.

Because our mark-recapture data were collected as part of a time series that spanned different parts of the growing season, incorporation of covariates including physical habitat variables and energetic inputs (e.g., prey availability, density-dependence) was intractable. Exploratory multiple regression analyses attempting to predict growth data with variables such as fish density, temperature and current velocity yielded no discernible pattern [33]. Continuous monitoring of these variables is costly and, when fish are marked and recaptured at different time points in the season, the measured growing interval can vary among individual fish. Although we did not measure food availability, increased production of stream invertebrates sometimes results from addition of ELJs [35, 74] and could be the mechanism behind the observed growth benefit in the ELJ-enhanced pools. Relative to shallow, faster-flowing habitats such as riffles, stream pools have slower current velocity and thus lower delivery rate of the drifting macroinvertebrates consumed by juvenile salmonids. Theoretical [75, 76] and empirical [20, 77, 78] studies indicate energetic trade-offs consistent with lower productivity in pools and that the cover offered by ELJs serves primarily in anti-predator vigilance [41]. Nevertheless, we have shown that ELJ-enhanced pools provide a growth benefit to fish and, although there can be weak effects of ELJs on macroinvertebrate production [74], exploration of this mechanism is warranted.

Our work has implications for the use of life cycle models to predict population-level responses of salmonids to restoration [5, 79], but there are also substantial uncertainties. These models examine changes in habitat capacity for fish, but previously such changes had not been detected in a nearby sub-basin [17]. We found positive growth in restored habitat, which is evidence of capacity increase, and provides strong support for the recent finding of a capacity increase using fish density in this system [73], but whether this is important for the entire life cycle remains unclear. Salmonid growth at one life stage is sometimes a strong predictor of

growth at a subsequent stage [65], but not always [80]. More broadly, observations of increased growth at a few sites of course does not mean that growth would be increased over larger areas.

Model fitting nevertheless allowed us to show that growth can be reliably quantified as a biological response to habitat quality differences, a rare accomplishment in fish restoration. Model fitting also provided at least partial confirmation that differences in growth between habitats are concordant with differences in relative density between habitats, and are evident despite higher overall density in restored habitat. It is also important to note that, in 2010, higher steelhead growth was not associated with higher abundance, emphasizing that there is more to restoration than abundance. Finally, the model strongly suggests that phenology and competition may be modulated by habitat restoration. Growth models may thus provide a tool of general usefulness in restoration ecology.

## Supporting information

**S1 Text.**
(TXT)

**S1 File.**
(ZIP)

## Acknowledgments

We thank Cascadia Conservation District, S. Eichler, R. Logan, A. Bushy (supported by the American Fisheries Society Hutton Junior Fisheries Biology Scholarship), K. Tackman, J. West, J. Novak, K. Sirianni, L. Flynn, K. Logan, J. Jorgensen, H. Porter, B. Forney, S. Letzing, R. Hosman, S. Kaech, N. Holt, K. Swieca, O. Graham, C. Skalisky, R. An and S. Claeson for field assistance and/or for assistance in data preparation during the various years of the study. Reviews of earlier drafts and helpful input were provided by C. Pfister, S. Carran, R. Flitcroft and A. Rosenberger. Ethical standards were evaluated and approved by S. Alexander, Program Manager, Land and Watershed Management Program, Pacific Northwest Research Station.

## Author Contributions

**Conceptualization:** Carlos M. Polivka, Greg Dwyer.

**Data curation:** Carlos M. Polivka.

**Formal analysis:** Carlos M. Polivka, Joseph R. Mihaljevic.

**Funding acquisition:** Carlos M. Polivka.

**Investigation:** Carlos M. Polivka.

**Methodology:** Carlos M. Polivka, Joseph R. Mihaljevic, Greg Dwyer.

**Project administration:** Carlos M. Polivka.

**Resources:** Carlos M. Polivka.

**Software:** Joseph R. Mihaljevic.

**Supervision:** Carlos M. Polivka.

**Writing – original draft:** Carlos M. Polivka, Joseph R. Mihaljevic, Greg Dwyer.

**Writing – review & editing:** Carlos M. Polivka.

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
