## [Decision Letter · Decision Letter 0]

11 Mar 2020

PONE-D-20-02574

Use of a mechanistic growth model in evaluating post-restoration habitat quality for juvenile salmonids

PLOS ONE

Dear Dr. Polivka,

Thank you for submitting your manuscript to PLOS ONE. After careful consideration, we feel that it has merit but does not fully meet PLOS ONE’s publication criteria as it currently stands. Therefore, we invite you to submit a revised version of the manuscript that addresses the points raised during the review process.

Academic Editor

You have received two reviews of your manuscript, which are generally positive.  Each has raised a few issues that must be addressed before this paper can be considered for publication in PlosOne.  Please address these issues in the revised manuscript, or in an accompanying letter.

We would appreciate receiving your revised manuscript by Apr 25 2020 11:59PM. To enhance the reproducibility of your results, we recommend that if applicable you deposit your laboratory protocols in protocols.io, where a protocol can be assigned its own identifier (DOI) such that it can be cited independently in the future. For instructions see: http://journals.plos.org/plosone/s/submission-guidelines#loc-laboratory-protocols

We look forward to receiving your revised manuscript.

Kind regards,

Maura (Gee) Geraldine Chapman, PhD DSc

Academic Editor

PLOS ONE

Additional Editor Comments (if provided):

Academic Editor

You have received two reviews of your manuscript, which are generally positive. Each has raised a few issues that must be addressed before this paper can be considered for publication in PlosOne. Please address these issues in the revised manuscript, or in an accompanying letter.

Journal Requirements:

"Early portions of this work (2009-2010) were funded by Bonneville Power Administration (Project No. 2003-017-00), and by the American Recovery and Re-investment Act enacted by President B. Obama. The later activities (2012-2016) were funded by the U.S. Bureau of Reclamation. JRM was funded by a US Department of Agriculture (USDA) National Institute of Food and Agriculture (NIFA) Postdoctoral Fellowship (2014-67012-22272)."

"Funding to the authors is described in the acknowledgements. The funders had no role in study design, data collection and analysis, decision to publish, or preparation of the manuscript."

3. Your ethics statement must appear in the Methods section of your manuscript. If your ethics statement is written in any section besides the Methods, please move it to the Methods section and delete it from any other section. Please also ensure that your ethics statement is included in your manuscript, as the ethics section of your online submission will not be published alongside your manuscript.

Reviewers' comments:

Reviewer's Responses to Questions

**Comments to the Author**

1. Is the manuscript technically sound, and do the data support the conclusions?

Reviewer #1: Yes

Reviewer #2: Yes

2. Has the statistical analysis been performed appropriately and rigorously? 

Reviewer #1: Yes

Reviewer #2: Yes

3. Have the authors made all data underlying the findings in their manuscript fully available?

Reviewer #1: Yes

Reviewer #2: Yes

4. Is the manuscript presented in an intelligible fashion and written in standard English?

Reviewer #1: Yes

Reviewer #2: Yes

5. Review Comments to the Author

Reviewer #1: This study is a great example of using novel methods to evaluate the age-old question of evaluating restoration effectiveness. I think measuring the growth effects, rather than changes in local abundance, is a great metric that is often not evaluated because other approaches have not worked well, or the mark-recapture data have been too difficult to obtain. Measures of changes in abundance are often discounted because it is assumed that restoration is merely drawing fish from other habitats with no appreciable benefit to the population. However, growth, as the authors mention has correlations with other important demographic parameters. In short, I think this manuscript illustrates a technique that others should and probably will take advantage of when evaluating restoration. For me, as someone unfamiliar with the Entiat River, the management implications are harder to wrap my head around. The authors state that the unrestored reach is nearby, and also has pools, but those pools are smaller.

I always struggle with studies of this type where the comparison is commonly to an unrestored area. I am always wanting to see comparison to a “reference” area, where instead of asking “does restoration improve growth compared to unrestored areas?” you might ask “Is the restoration providing similar growth as a natural wood jam produced pool?” These questions are different but we tend to only see the former, not the latter, and I don’t know why. Maybe there is no natural wood recruitment in the system?

Does the unrestored reach also have wood, or are the pools different in other aspects besides their size? I suppose I was expecting to see this aspect introduced as a testable hypothesis.

Were you expecting that growth would be enhanced by the new pools? If so, why?

Might there be a more diverse invertebrate community? Is there more cover allowing for less vigilance among individuals?

You mention that the densities are similar among restored and unrestored reaches. Is there possibly then, another aspect of the allometry of pool size that provides thresholds of benefit at similar volumetric density? I would think so. It does appear that the interspecific interactions are very important as the authors suggest, and there is a clear residence time component. I think what I am left wondering is if the unrestored reach had larger pools, would we see a difference, or is it the wood, irrespective of pool size that provides the benefit? You mention that work with specific environmental covariates yielded nothing conclusive, but form a restoration perspective, is it the pools or the wood or does that matter because you cannot get large pools without the wood?

Unrestored pools are smaller, and there are no recapture records of Chinook I those habitats. Could it be that the unrestored habitat is really representative of a smaller transient class of fish that is moving downstream, rather than indicative of residence in that habitat?

L88: is this a reference? “beechie2005classification”

Figure 1 could include a little more context. Can you provide a scale bar?

Figure 3. Is this just a sample “hypothetical” plot? Why not label the arrow with R-hat to make it work with the caption.

Reviewer #2: Review of PONE-D-02574

This paper presents a mechanistic growth model used to evaluate the ecological effectiveness of river habitat enhancement on threatened juvenile salmonid populations in Washington state. Most habitat restoration/enhancement studies rely on density estimates to determine whether a conservation action achieved its objective even though density is a misleading indicator of habitat quality. The authors applied the model using growth data collected from mark-recapture of juvenile fish in a river reach where large wood was added and compared these data to an unrestored reach. They also compared growth results to density estimates to determine congruence in how they reflected the response to habitat enhancement. Model parameters not only described the rate of individual growth but also the timing of increases. The model was also spatially explicit accounting for habitat differences. The authors suggest that these characteristics make it widely applicable to different taxa.

General comments: I found this article well-written and logically presented. Moreover, the topic is timely as billions of dollars have been spent to enhance habitat degraded by human activities but approaches to assess the effectiveness of these actions are limited to mostly monitoring changes in density, which is not a reliable indicator of habitat quality.

I would suggest the authors use “habitat enhancement” rather than habitat restoration as the project they evaluated did not restore the process of wood loading to river channels. Instead, the project enhanced wood abundance in the river.

Specific comments:

Line 36: This is the first time we learn about salmonids, which are the main players in the paper. More details on what they are and why they are of interest is needed. Also, I would suggest introducing them earlier in the introduction.

Figure 1. I’m not sure how much the map adds to the paper.

Fish marking: What about fish movement? This is an open system where fish are moving among habitats. The authors need to address this complication to their story.

Local density: Why is local density not part of the model?

Line 161: I would like to see an explicit definition of �� (alpha) here. We don’t learn what it is in detail until later in the paper.

Line 388-389: Provide references for this statement.

Any ideas of why there were annual differences in response (e.g., Figure 8)?

Any ideas of why there were annual differences in response (e.g., Figure 8)?

6. PLOS authors have the option to publish the peer review history of their article (what does this mean?). If published, this will include your full peer review and any attached files.

Reviewer #1: No

Reviewer #2: Yes: Peter M. Kiffney

---

## [Editor Report · Decision Letter 1]

19 May 2020

Use of a mechanistic growth model in evaluating post-restoration habitat quality for juvenile salmonids

PONE-D-20-02574R1

Dear Dr. Polivka,

We are pleased to inform you that your manuscript has been judged scientifically suitable for publication and will be formally accepted for publication once it complies with all outstanding technical requirements.

With kind regards,

Maura (Gee) Geraldine Chapman, PhD DSc

Academic Editor

PLOS ONE
---

## [Editor Report · Acceptance letter]

15 Jun 2020

PONE-D-20-02574R1 

Use of a mechanistic growth model in evaluating post-restoration habitat quality for juvenile salmonids 

Dear Dr. Polivka:

I'm pleased to inform you that your manuscript has been deemed suitable for publication in PLOS ONE. Congratulations! Your manuscript is now with our production department. 

Kind regards, 

on behalf of

Professor Maura (Gee) Geraldine Chapman 

Academic Editor

PLOS ONE